# Mirror Descent View for Neural Network Quantization

## Abstract

Quantizing large Neural Networks (NN) while maintaining the performance is highly desirable for resource-limited devices due to reduced memory and time complexity. NN quantization is usually formulated as a constrained optimization problem and optimized via a modified version of gradient descent. In this work, by interpreting the continuous parameters (unconstrained) as the dual of the quantized ones, we introduce a Mirror Descent (MD) framework (Bubeck (2015)) for NN quantization. Specifically, we provide conditions on the projections (*i.e.*, mapping from continuous to quantized ones) which would enable us to derive valid mirror maps and in turn the respective MD updates. Furthermore, we discuss a numerically stable implementation of MD by storing an additional set of auxiliary dual variables (unconstrained). This update is strikingly analogous to the popular Straight Through Estimator (STE) based method which is typically viewed as a "trick" to avoid vanishing gradients issue but here we show that it is an implementation method for MD for certain projections. Our experiments on standard classification datasets (CIFAR-10/100, TinyImageNet) with convolutional and residual architectures show that our MD variants obtain fully-quantized networks with accuracies very close to the floating-point networks.

## 1 Introduction

Despite the success of deep neural networks in various domains, their excessive computational and memory requirements limit their practical usability for real-time applications or in resource-limited devices. Quantization is a prominent technique for network compression, where the objective is to learn a network while restricting the parameters to take values from a small discrete set (usually binary). This leads to a dramatic reduction in memory (a factor of 32 for binary quantization) and inference time – as it enables specialized implementation using bit operations.

Neural Network (NN) quantization is usually formulated as a constrained optimization problem $\min_{\mathbf{x} \in \mathcal{X}} f(\mathbf{x})$, where $f(\cdot)$ denotes the loss function by abstracting out the dependency on the dataset and $\mathcal{X} \subset \mathbb{R}^r$ denotes the set of all possible quantized solutions. Majority of the works in the literature (Hubara et al. (2017); Yin et al. (2018); Ajanthan et al. (2019)) convert this into an unconstrained problem by introducing auxiliary variables ($\tilde{\mathbf{x}}$) and optimize via (stochastic) gradient descent. Specifically, the objective and the update step take the following form:

$$\min_{\tilde{\mathbf{x}} \in \mathbb{R}^r} f(P(\tilde{\mathbf{x}})) , \qquad \tilde{\mathbf{x}}^{k+1} = \tilde{\mathbf{x}}^k - \eta \, \nabla_{\tilde{\mathbf{x}}} f(P(\tilde{\mathbf{x}}))|_{\tilde{\mathbf{x}} = \tilde{\mathbf{x}}^k} , \qquad (1)$$

where $P : \mathbb{R}^r \to \mathcal{X}$ is a mapping from the unconstrained space to the quantized space (sometimes called projection) and $\eta > 0$ is the learning rate. In cases where the mapping $P$ is not differentiable, a suitable approximation is employed (Hubara et al. (2017)).

In this work, by noting that the well-known Mirror Descent (MD) algorithm, widely used for online convex optimization (Bubeck (2015)), provides a theoretical framework to perform gradient descent in the unconstrained space (dual space, $\mathbb{R}^r$) with gradients computed in the quantized space (primal space, $\mathcal{X}$), we introduce an MD framework for NN quantization. In essence, MD extends gradient descent to non-Euclidean spaces where Euclidean projection is replaced with a more general projection defined based on the associated distance metric. Briefly, the key ingredient of MD is a concept called *mirror map* which defines both the mapping between primal and dual spaces and the exact form of

the projection. Specifically, in this work, by observing $P$ in Eq. (1) as a mapping from dual space to the primal space, we analytically derive corresponding mirror maps under certain conditions on $P$. This enables us to derive different variants of the MD algorithm useful for NN quantization.

Furthermore, as MD is often found to be numerically unstable (Hsieh et al. (2018)), we discuss a numerically stable implementation of MD by storing an additional set of auxiliary variables similar to the existing methods. As will be shown later, this update is strikingly analogous to the popular Straight Through Estimator (STE) based gradient method (Hubara et al. (2017); Bai et al. (2019)) which is typically viewed as a "trick" to avoid vanishing gradients issue but here we show that it is an implementation method for MD under certain conditions on the mapping $P$. We believe this connection sheds some light on the practical effectiveness of STE.

We evaluate the merits of our MD variants on CIFAR-10/100 and TinyImageNet classification datasets with convolutional and residual architectures. Our experiments show that the quantized networks obtained by the MD variants yield accuracies very close to the floating-point counterparts while outperforming directly comparable baselines. Finally, we would like to emphasize that even though our formulation does not necessarily extend the theory of MD, we believe showing MD as a suitable framework for NN quantization with superior empirical performance opens up new ways of designing MD-inspired update rules for NNs.

## 2 PRELIMINARIES

We first provide some background on the MD algorithm and NN quantization. Then we discuss the link between them and provide our MD framework for NN quantization.

### 2.1 MIRROR DESCENT

The Mirror Descent (MD) algorithm is first introduced in (Nemirovsky & Yudin (1983)) and it has been extensively studied in the convex optimization literature ever since. In this section we provide a brief overview and we refer the interested reader to Chapter 4 of (Bubeck (2015)). In the context of MD, we consider a problem of the form:

$$\min_{\mathbf{x} \in \mathcal{X}} f(\mathbf{x}) \,, \tag{2}$$

where $f : \mathcal{X} \to \mathbb{R}$ is a convex function and $\mathcal{X} \subset \mathbb{R}^r$ is a compact convex set. The main concept of MD is to extend gradient descent to a more general non-Euclidean space (Banach space[1]), thus overcoming the dependency of gradient descent on the Euclidean geometry. The motivation for this generalization is that one might be able to exploit the geometry of the space to optimize much more efficiently. One such example is the simplex constrained optimization where MD converges at a much faster rate than the standard Projected Gradient Descent (PGD).

To this end, since the gradients lie in the dual space, optimization is performed by first mapping the primal point $\mathbf{x}^k \in \mathcal{B}$ (quantized space, $\mathcal{X}$) to the dual space $\mathcal{B}^*$ (unconstrained space, $\mathbb{R}^r$), then performing gradient descent in the dual space, and finally mapping back the resulting point to the primal space $\mathcal{B}$. If the new point $\mathbf{x}^{k+1}$ lie outside of the constraint set $\mathcal{X} \subset \mathcal{B}$, it is projected to the set $\mathcal{X}$. Both the primal/dual mapping and the projection are determined by the *mirror map*. Specifically, the gradient of the mirror map defines the mapping from primal to dual and the projection is done via the Bregman divergence of the mirror map. We first provide the definitions for mirror map and Bregman divergence and then turn to the MD updates.

**Definition 2.1** (Mirror map). Let $\mathcal{C} \subset \mathbb{R}^r$ be a convex open set such that $\mathcal{X} \subset \bar{\mathcal{C}}$ ($\bar{\mathcal{C}}$ denotes the closure of set $\mathcal{C}$) and $\mathcal{X} \cap \mathcal{C} \neq \emptyset$. Then, $\Phi : \mathcal{C} \to \mathbb{R}$ is a mirror map if it satisfies:

1. $\Phi$ is strictly convex and differentiable.
2. $\nabla\Phi(\mathcal{C}) = \mathbb{R}^r$, *i.e.*, $\nabla\Phi$ takes all possible values in $\mathbb{R}^r$.
3. $\lim_{\mathbf{x} \to \partial\mathcal{C}} \|\nabla\Phi(\mathbf{x})\| = \infty$ ($\partial\mathcal{C}$ denotes the boundary of $\mathcal{C}$), *i.e.*, $\nabla\Phi$ diverges on the boundary of $\mathcal{C}$.

**Definition 2.2** (Bregman divergence). Let $\Phi : \mathcal{C} \to \mathbb{R}$ be a continuously differentiable, strictly convex function defined on a convex set $\mathcal{C}$. The Bregman divergence associated with $\Phi$ for points $\mathbf{p}, \mathbf{q} \in \mathcal{C}$ is the difference between the value of $\Phi$ at point $\mathbf{p}$ and the value of the first-order Taylor expansion of $\Phi$ around point $\mathbf{q}$ evaluated at point $\mathbf{p}$, *i.e.*,

$$D_\Phi(\mathbf{p}, \mathbf{q}) = \Phi(\mathbf{p}) - \Phi(\mathbf{q}) - \langle\nabla\Phi(\mathbf{q}), \mathbf{p} - \mathbf{q}\rangle \,. \tag{3}$$

---

[1] A Banach space is a complete normed vector space where the norm is not necessarily derived from an inner product.

Notice, $D_\Phi(\mathbf{p}, \mathbf{q}) \geq 0$ with $D_\Phi(\mathbf{p}, \mathbf{p}) = 0$, and $D_\Phi(\mathbf{p}, \mathbf{q})$ is convex on $\mathbf{p}$.

Now we are ready to provide the mirror descent strategy based on the mirror map $\Phi$. Let $\mathbf{x}^0 \in \operatorname{argmin}_{\mathbf{x} \in \mathcal{X} \cap \mathcal{C}} \Phi(\mathbf{x})$ be the initial point. Then, for iteration $k \geq 0$ and step size $\eta > 0$, the update of the MD algorithm can be written as:

$$\nabla\Phi(\mathbf{y}^{k+1}) = \nabla\Phi(\mathbf{x}^k) - \eta\,\mathbf{g}^k\,, \qquad \text{where } \mathbf{g}^k \in \partial f(\mathbf{x}^k) \text{ and } \mathbf{y}^{k+1} \in \mathcal{C}\,, \qquad (4)$$
$$\mathbf{x}^{k+1} = \operatorname*{argmin}_{\mathbf{x} \in \mathcal{X} \cap \mathcal{C}} D_\Phi(\mathbf{x}, \mathbf{y}^{k+1})\,.$$

Note that, in Eq. (4), the gradient $\mathbf{g}^k$ is computed at $\mathbf{x}^k \in \mathcal{X} \cap \mathcal{C}$ (solution space) but the gradient descent is performed in $\mathbb{R}^r$ (unconstrained dual space). Moreover, by simple algebraic manipulation, it is easy to show that the above MD update (4) can be compactly written in a proximal form where the Bregman divergence of the mirror map becomes the proximal term (Beck & Teboulle (2003)):

$$\mathbf{x}^{k+1} = \operatorname*{argmin}_{\mathbf{x} \in \mathcal{X} \cap \mathcal{C}} \langle \eta\,\mathbf{g}^k, \mathbf{x} \rangle + D_\Phi(\mathbf{x}, \mathbf{x}^k)\,. \qquad (5)$$

Note, if $\Phi(\mathbf{x}) = \frac{1}{2}\|\mathbf{x}\|_2^2$, then $D_\Phi(\mathbf{x}, \mathbf{x}^k) = \frac{1}{2}\|\mathbf{x} - \mathbf{x}^k\|_2^2$, which when plugged back to the above problem and optimized for $\mathbf{x}$, leads to exactly the same update rule as that of PGD. However, MD allows us to choose various forms of $\Phi$ depending on the problem at hand.

## 2.2 NEURAL NETWORK QUANTIZATION

Neural Network (NN) quantization amounts to training networks with parameters restricted to a small discrete set representing the quantization levels. Here we review two constrained optimization formulations for NN quantization: 1) directly constrain each parameter to be in the discrete set; and 2) optimize the probability of each parameter taking a label from the set of quantization levels.

### 2.2.1 PARAMETER SPACE FORMULATION

Given a dataset $\mathcal{D} = \{\mathbf{x}_i, \mathbf{y}_i\}_{i=1}^n$, NN quantization can be written as:

$$\min_{\mathbf{w} \in \mathcal{Q}^m} L(\mathbf{w}; \mathcal{D}) := \frac{1}{n}\sum_{i=1}^n \ell(\mathbf{w}; (\mathbf{x}_i, \mathbf{y}_i))\,. \qquad (6)$$

Here, $\ell(\cdot)$ denotes the input-output mapping composed with a standard loss function (*e.g.*, cross-entropy loss), $\mathbf{w}$ is the $m$ dimensional parameter vector, and $\mathcal{Q}$ with $|\mathcal{Q}| = d$ is a predefined discrete set representing quantization levels (*e.g.*, $\mathcal{Q} = \{-1, 1\}$ or $\mathcal{Q} = \{-1, 0, 1\}$).

The approaches that directly optimize in the parameter space include BinaryConnect (BC) (Courbariaux et al. (2015)) and its variants (Hubara et al. (2017); Rastegari et al. (2016)), where the constraint set is discrete. In contrast, recent approaches (Bai et al. (2019); Yin et al. (2018)) relax this constraint set to be its convex hull:

$$\operatorname{conv}(\mathcal{Q}^m) = [q_{\min}, q_{\max}]^m\,, \qquad (7)$$

where $q_{\min}$ and $q_{\max}$ represent the minimum and maximum quantization levels, respectively. In this case, a quantized solution is obtained by gradually increasing an annealing hyperparameter.

### 2.2.2 LIFTED PROBABILITY SPACE FORMULATION

Another formulation is based on the Markov Random Field (MRF) perspective to NN quantization recently studied in (Ajanthan et al. (2019)). It treats Eq. (6) as a *discrete labelling problem* and introduces indicator variables $u_{j:\lambda} \in \{0, 1\}$ for each parameter $w_j$ where $j \in \{1, \ldots, m\}$ such that $u_{j:\lambda} = 1$ if and only if $w_j = \lambda \in \mathcal{Q}$. For convenience, by denoting the vector of quantization levels as $\mathbf{q}$, a parameter vector $\mathbf{w} \in \mathcal{Q}^m$ can be written in a matrix vector product as:

$$\mathbf{w} = \mathbf{u}\mathbf{q}\,, \quad \text{where} \quad \mathbf{u} \in \mathcal{V}^m = \left\{ \mathbf{u} \,\middle|\, \begin{array}{ll} \sum_\lambda u_{j:\lambda} = 1, & \forall j \\ u_{j:\lambda} \in \{0, 1\}, & \forall j, \lambda \end{array} \right\}\,. \qquad (8)$$

Here, $\mathbf{u}$ is a $m \times d$ matrix (*i.e.*, each row $\mathbf{u}_j = \{u_{j:\lambda} \mid \lambda \in \mathcal{Q}\}$), and $\mathbf{q}$ is a column vector of dimension $d$. Note that, $\mathbf{u} \in \mathcal{V}^m$ is an overparametrized (*i.e.*, lifted) representation of $\mathbf{w} \in \mathcal{Q}^m$. Now, similar to the relaxation in the parameter space, one can relax the binary constraint in $\mathcal{V}^m$ to form its convex hull:

$$\Delta^m = \operatorname{conv}(\mathcal{V}^m) = \left\{ \mathbf{u} \,\middle|\, \begin{array}{ll} \sum_\lambda u_{j:\lambda} = 1, & \forall j \\ u_{j:\lambda} \geq 0, & \forall j, \lambda \end{array} \right\}\,. \qquad (9)$$

The set $\Delta^m$ is in fact the Cartesian product of the standard $(d-1)$-probability simplexes embedded in $\mathbb{R}^d$. Therefore, for a feasible point $\mathbf{u} \in \Delta^m$, the vector $\mathbf{u}_j$ for each $j$ ($j$-th row of matrix $\mathbf{u}$) belongs to the probability simplex $\Delta$. Hence, we can interpret the value $u_{j:\lambda}$ as the probability of assigning the discrete label $\lambda$ to the weight $w_j$. This relaxed optimization can then be written as:

$$\min_{\mathbf{u} \in \Delta^m} L(\mathbf{uq}; \mathcal{D}) := \frac{1}{n} \sum_{i=1}^{n} \ell(\mathbf{uq}; (\mathbf{x}_i, \mathbf{y}_i)) \,. \tag{10}$$

In fact, this can be interpreted as finding a probability distribution $\mathbf{u} \in \Delta^m$ such that the cost $L(\mathbf{u})$ is minimized. Note that, the relaxation of $\mathbf{u}$ from $\mathcal{V}^m$ to $\Delta^m$ translates into relaxing $\mathbf{w}$ from $\mathcal{Q}^m$ to the convex region $\mathrm{conv}(\mathcal{Q}^m)$. Even in this case, a discrete solution $\mathbf{u} \in \mathcal{V}^m$ can be enforced via an annealing hyperparameter or using rounding schemes.

## 3 MIRROR DESCENT FRAMEWORK FOR NETWORK QUANTIZATION

Before introducing the MD formulation, we first write NN quantization as a single objective unifying (6) and (10) as:

$$\min_{\mathbf{x} \in \mathcal{X}} f(\mathbf{x}) \,, \tag{11}$$

where $f(\cdot)$ denotes the loss function by abstracting out the dependency on the dataset $\mathcal{D}$, and $\mathcal{X}$ denotes the constraint set ($\mathcal{Q}^m$ or $\mathcal{V}^m$ depending on the formulation). Note that, as discussed in Sec. 2.2, many recent NN quantization methods optimize over the convex hull of the constraint set. Following this, we consider the solution space $\mathcal{X}$ in Eq. (11) to be convex and compact. To employ MD, we need to choose a mirror map (refer Definition 2.1) suitable for the problem at hand. In fact, as discussed in Sec. 2.1, mirror map is the core component of an MD algorithm which determines the effectiveness of the resulting MD updates. However, there is no straightforward approach to obtain a mirror map for a given constrained optimization problem, except in certain special cases.

To this end, we observe that the usual approach to optimize the above constrained problem is via a version of projected gradient descent, where the projection is the mapping from the unconstrained auxiliary variables (high-precision) to the quantized space $\mathcal{X}$. Now, noting the analogy between the purpose of the projection operator and the mirror maps in the MD formulation, we intend to derive the mirror map analogous to a given projection. Precisely, we prove that if the projection is invertible and strictly monotone, a valid mirror map can be derived from the projection itself. This does not necessarily extend the theory of MD as finding a strictly monotone map is as hard as finding the mirror map itself. However, this derivation is interesting as it connects existing PGD type algorithms to their corresponding MD variants. For completeness, we now state it as a theorem for the case $\mathcal{X} \subset \mathbb{R}$ and the multidimensional case can be easily proved with an additional assumption that the vector field $P^{-1}(\mathbf{x})$ is conservative.

**Theorem 3.1.** *Let $\mathcal{X}$ be a compact convex set and $P : \mathbb{R} \to \mathcal{C}$ be an invertible function where $\mathcal{C} \subset \mathbb{R}$ is a convex open set such that $\mathcal{X} = \bar{\mathcal{C}}$ ($\bar{\mathcal{C}}$ denotes the closure of $\mathcal{C}$). Now, if*

*1. $P$ is strictly monotonically increasing.*
*2. $\lim_{x \to \partial \mathcal{C}} \|P^{-1}(x)\| = \infty$ ($\partial \mathcal{C}$ denotes the boundary of $\mathcal{C}$).*

*Then, $\Phi(x) = \int_{x_0}^{x} P^{-1}(y)dy$ is a valid mirror map.*

*Proof.* This can be proved by noting that $\nabla \Phi(x) = P^{-1}(x)$. Please refer to Appendix A. □

The MD update based on the mirror map derived from a given projection is illustrated in Fig. 1. Note that, to employ MD to the problem (11), in theory, any mirror map satisfying Definition 2.1 whose domain (the closure of the domain) is a superset of the constraint set $\mathcal{X}$ can be chosen. However, the above theorem provides a method to derive only a subset of all applicable mirror maps, where the closure of the domain of mirror maps is exactly equal to the constraint set $\mathcal{X}$.

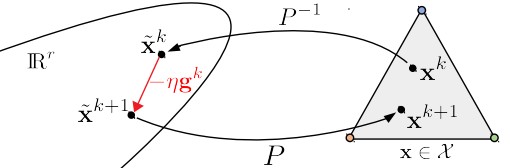

Figure 1: MD *formulation where mirror map is derived from the projection $P$. Note, $\mathbf{g}^k$ is computed in the primal space ($\mathcal{X}$) but it is directly used to update the auxiliary variables in the dual space.*

We now give some example projections useful for NN quantization ($\tanh$ for $\mathbf{w}$-space and $\mathrm{softmax}$ for $\mathbf{u}$-space) and derive their corresponding mirror maps. Given mirror maps, the MD updates are straightforward based on Eq. (5). Even though we consider differentiable projections, Theorem 3.1

does not require the projection to be differentiable. For the rest of the section, we assume $m = 1$, *i.e.*, consider projections that are independent for each $j \in \{1, \ldots, m\}$.

**Example 3.1** (**w**-space, binary, $\tanh$). Consider the $\tanh$ function, which projects a real value to the interval $[-1, 1]$:

$$w = P(\tilde{w}) := \tanh(\beta\tilde{w}) = \frac{\exp(2\beta\tilde{w}) - 1}{\exp(2\beta\tilde{w}) + 1} , \qquad (12)$$

where $\beta > 0$ is the annealing hyperparameter and when $\beta \to \infty$, $\tanh$ approaches the step function. The inverse of the $\tanh$ is:

$$P^{-1}(w) = \frac{1}{\beta} \tanh^{-1}(w) = \frac{1}{2\beta} \log \frac{1+w}{1-w} . \qquad (13)$$

Note that, $P^{-1}$ is monotonically increasing for a fixed $\beta$. Correspondingly, the mirror map from Theorem 3.1 can be written as:

$$\Phi(w) = \int P^{-1}(w)dw = \frac{1}{2\beta}\big[(1+w)\log(1+w) + (1-w)\log(1-w)\big] . \qquad (14)$$

Here, the constant from the integration is ignored. It can be easily verified that $\Phi(w)$ is in fact a valid mirror map. The projection, its inverse and the corresponding mirror map are illustrated in Fig. 2a. Consequently, the resulting MD update (5) takes the following form:

$$w^{k+1} = \operatorname*{argmin}_{w \in (-1,1)} \langle \eta\, g^k, w \rangle + D_\Phi(w, w^k) = \frac{\frac{1+w^k}{1-w^k} \exp(-2\beta\eta g^k) - 1}{\frac{1+w^k}{1-w^k} \exp(-2\beta\eta g^k) + 1} . \qquad (15)$$

The update formula is derived using the KKT conditions (Boyd & Vandenberghe (2009)). For the detailed derivation please refer to Appendix B. A similar derivation can also be performed for the sigmoid function, where $\bar{\mathcal{C}} = \mathcal{X} = [0, 1]$. Note that the sign function has been used for binary quantization in (Courbariaux et al. (2015)) and $\tanh$ can be used as a soft version of sign function as pointed out by (Zhang et al. (2015)). Mirror map corresponding to $\tanh$ is used for online linear optimization in (Bubeck et al. (2012)) but here we use it for NN quantization.

**Example 3.2** (**u**-space, multi-label, softmax). Now we consider the softmax projection used in Proximal Mean-Field (PMF) (Ajanthan et al. (2019)) to optimize in the lifted probability space. In this case, the projection is defined as $P(\tilde{\mathbf{u}}) := \text{softmax}(\beta\tilde{\mathbf{u}})$ where $P : \mathbb{R}^d \to \mathcal{C}$ with $\bar{\mathcal{C}} = \mathcal{X} = \Delta$. Here $\Delta$ is the $(d-1)$-dimensional probability simplex and $|\mathcal{Q}| = d$. Note that the softmax projection is not invertible as it is a many-to-one mapping. In particular, it is invariant to translation, *i.e.*,

$$\mathbf{u} = \text{softmax}(\tilde{\mathbf{u}} + c\mathbf{1}) = \text{softmax}(\tilde{\mathbf{u}}) , \qquad \text{where} \quad u_\lambda = \frac{\exp(\tilde{u}_\lambda)}{\sum_{\mu \in \mathcal{Q}} \exp(\tilde{u}_\mu)} , \qquad (16)$$

for any scalar $c \in \mathbb{R}$ (**1** denotes a vector of all ones). Therefore, the softmax projection does not satisfy Theorem 3.1. However, one could define the inverse of softmax as follows: given $\mathbf{u} \in \Delta$, find a unique point $\tilde{\mathbf{v}} = \tilde{\mathbf{u}} + c\mathbf{1}$, for a particular scalar $c$, such that $\mathbf{u} = \text{softmax}(\tilde{\mathbf{v}})$. Now, by choosing $c = -\log(\sum_{\mu=\mathcal{Q}} \exp(\tilde{u}_\mu))$, softmax can be written as:

$$\mathbf{u} = \text{softmax}(\tilde{\mathbf{v}}) , \qquad \text{where} \quad u_\lambda = \exp(\tilde{v}_\lambda) , \quad \forall\, \lambda \in \mathcal{Q} . \qquad (17)$$

Now, the inverse of the projection can be written as:

$$\tilde{\mathbf{v}} = P^{-1}(\mathbf{u}) = \frac{1}{\beta} \text{softmax}^{-1}(\mathbf{u}) , \qquad \text{where} \quad \tilde{v}_\lambda = \frac{1}{\beta} \log(u_\lambda) , \qquad \forall\, \lambda . \qquad (18)$$

Indeed, $\log$ is a monotonically increasing function and from Theorem 3.1, by summing the integrals, the mirror map can be written as:

$$\Phi(\mathbf{u}) = \frac{1}{\beta} \left[ \sum_\lambda u_\lambda \log(u_\lambda) - u_\lambda \right] = -\frac{1}{\beta} H(\mathbf{u}) - 1/\beta . \qquad (19)$$

Here, $\sum_\lambda u_\lambda = 1$ as $\mathbf{u} \in \Delta$, and $H(\mathbf{u})$ is the entropy. Interestingly, as the mirror map in this case is the negative entropy (up to a constant), the MD update leads to the well-known Exponentiated

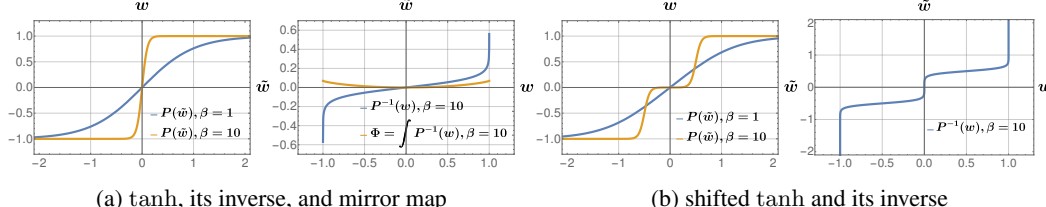

(a) tanh, its inverse, and mirror map           (b) shifted tanh and its inverse

Figure 2: *Plots of* tanh *and shifted* tanh *projections, and their inverses corresponding to the* tanh *projection. Note that, the inverses are monotonically increasing and the mirror map is strictly convex. Moreover, when $\beta \to \infty$, the projections approaches their respective hard versions.*

Gradient Descent (EGD) (or Entropic Descent Algorithm (EDA)) (Beck & Teboulle (2003); Bubeck (2015)). Consequently, the update takes the following form:

$$u_\lambda^{k+1} = \frac{u_\lambda^k \, \exp(-\beta \eta g_\lambda^k)}{\sum_{\mu \in \mathcal{Q}} \, u_\mu^k \, \exp(-\beta \eta g_\mu^k)} \quad \forall \, \lambda \, . \tag{20}$$

The derivation follows the same approach as in the tanh case above. It is interesting to note that the MD variant of softmax is equivalent to the well-known EGD. Notice, the authors of PMF hinted that PMF is related to EGD but here we have clearly showed that the MD variant of PMF under the above reparametrization (17) is exactly EGD.

**Example 3.3** (w-space, multi-label, shifted tanh). Note that, similar to softmax, we wish to extend the tanh projection beyond binary. The idea is to use a function that is an addition of multiple shifted tanh functions. To this end, as an example we consider ternary quantization, with $\mathcal{Q} = \{-1, 0, 1\}$ and define our shifted tanh projection $P : \mathbb{R} \to \mathcal{C}$ as:

$$w = P(\tilde{w}) = \frac{1}{2}\Big[ \tanh\left(\beta(\tilde{w} + 0.5)\right) + \tanh\left(\beta(\tilde{w} - 0.5)\right) \Big] \, , \tag{21}$$

where $\beta > 0$ and $w = \bar{\mathcal{C}} = \mathcal{X} = [-1, 1]$. When $\beta \to \infty$, $P$ approaches a stepwise function with inflection points at $-0.5$ and $0.5$ (here, $\pm 0.5$ is chosen heuristically), meaning $w$ move towards one of the quantization levels in the set $\mathcal{Q}$. This behaviour together with its inverse is illustrated in Fig. 2b. Now, one could potentially find the functional form of $P^{-1}$ and analytically derive the mirror map corresponding to this projection. Note that, while Theorem 3.1 provides an analytical method to derive mirror maps, in some cases such as the above, the exact form of mirror map and the MD update might be nontrivial. In such cases, as will be shown subsequently, the MD update can be easily implemented by storing an additional set of auxiliary variables $\tilde{w}$.

**Effect of Annealing.** Note that, to ensure a discrete solution, projection $P$ is parametrized by a scalar $\beta$ and it is annealed throughout the optimization. This annealing hyperparameter translates into a time varying mirror map (refer Eqs. (14) and (19)) in our case. Such an adaptive mirror map gradually constrains the solution space $\mathcal{X}$ to its boundary and in the limit enforces a quantized solution. Since this adaptive behaviour can affect the convergence of the algorithm, in our implementation $\beta$ is capped at an arbitrarily chosen maximum value, and empirically, the algorithm converges to fully quantized solutions in all tested cases. We leave the theoretical analysis of annealing for future work.

**Numerically Stable form of Mirror Descent.** We showed a few examples of valid projections, their corresponding mirror maps, and the final MD updates. Even though, in theory, these updates can be used directly, they are sometimes numerically unstable due to the operations involving multiple logarithms, exponentials and divisions (Hsieh et al. (2018)). To this end, we provide a numerically stable way of performing MD by storing an additional set of auxiliary parameters during training.

A careful look at the Fig. 1 suggests that the MD update with the mirror map derived from Theorem 3.1 can be performed by storing auxiliary variables $\tilde{\mathbf{x}} = P^{-1}(\mathbf{x})$. In fact, once the auxiliary variable $\tilde{\mathbf{x}}^k$ is updated using gradient $\mathbf{g}^k$, it is directly mapped back to the constraint set $\mathcal{X}$ via the projection. This is mainly because of the fact that the domain of the mirror maps derived based on the Theorem 3.1 are exactly the same as the constraint set. Formally, with this additional set of variables, one can write the MD update (4) corresponding to the projection $P$ as:

$$\tilde{\mathbf{x}}^{k+1} = \tilde{\mathbf{x}}^k - \eta \, \mathbf{g}^k \, , \qquad \text{update in the dual space} \tag{22}$$

$$\mathbf{x}^{k+1} = P(\tilde{\mathbf{x}}^{k+1}) \in \mathcal{X} \, , \quad \text{projection to the primal space}$$

where $\eta > 0$, $\mathbf{g}^k \in \partial f(\mathbf{x}^k)$ and $\tilde{\mathbf{x}}^k = P^{-1}(\mathbf{x}^k)$. Experimentally we observed these updates to show stable behaviour and performed remarkably well for both the $\mathrm{tanh}$ and $\mathrm{softmax}$.

Note, above updates can be seen as optimizing the function $f(P(\tilde{\mathbf{x}}))$ using gradient descent where the gradient through the projection (*i.e.*, Jacobian) $J_P = \partial P(\tilde{\mathbf{x}})/\partial \tilde{\mathbf{x}}$ is replaced with the identity matrix. This is exactly the same as the Straight Through Estimator (STE) for NN quantization (following the nomenclature of (Bai et al. (2019); Yin et al. (2018))). Despite being a crude approximation, STE has shown to be highly effective for NN quantization with various network architectures and datasets (Yin et al. (2018); Zhou et al. (2016)). However, a solid understanding of the effectiveness of STE is lacking in the literature except for its convergence analysis in certain special cases (Li et al. (2017); Yin et al. (2019)). In this work, by showing STE based gradient descent as an implementation method of MD under certain conditions on the projection, we provide a justification on the effectiveness of STE. Besides, as shown in Example 3.3, in cases where the MD formulation is nontrivial, the STE based implementation can be used. The pseudocodes of original and numerically stable versions of our MD algorithm for $\mathrm{tanh}$ are presented in Appendix B.

**Comparison against ProxQuant.** The connection between the dual averaging version of MD and STE was recently hinted in ProxQuant (PQ) (Bai et al. (2019)). However, no analysis of whether an analogous mirror map exists to the given projection is provided and their final algorithm is not based on MD. In particular, following our notation, the final update equation of PQ can be written as:

$$\tilde{\mathbf{x}}^{k+1} = \mathbf{x}^k - \eta\,\mathbf{g}^k\,, \qquad \text{assumes } \mathbf{x}^k \text{ and } \mathbf{g}^k \text{ are in the same space} \tag{23}$$

$$\mathbf{x}^{k+1} = \mathrm{prox}(\tilde{\mathbf{x}}^{k+1})\,, \qquad \mathrm{prox} : \mathbb{R}^r \to \mathbb{R}^r \text{ is the proximal mapping defined in (Bai et al. (2019))}$$

where $\eta > 0$, and $\mathbf{g}^k \in \partial f(\mathbf{x}^k)$. Note that, as opposed to MD (refer to Eq. (22)), PQ assumes the point $\mathbf{x}^k$ and gradient $\mathbf{g}^k$ are in the same space. Then only the formula $\mathbf{x}^k - \eta\,\mathbf{g}^k$ is valid. This would only be true for the Euclidean space. However, as discussed in Sec. 2.1, MD allows gradient descent to be performed on a more general non-Euclidean space by first mapping the primal point $\mathbf{x}^k$ to a point $\tilde{\mathbf{x}}^k$ in the dual space via the mirror map. Such an ability enabled theoretical and practical research on MD for the past three decades.

**Convergence of MD in the Nonconvex Setting.** We would like to point out that MD is originally developed for convex optimization, however, in this paper we directly apply MD to NN quantization where the loss is highly nonconvex and gradient estimates are stochastic, and empirically evaluate its convergence behaviour and performance. Theoretical analysis of MD for nonconvex, stochastic setting is an active research area (Zhou et al. (2017a;b)) and MD has been recently shown to converge in the nonconvex stochastic setting under certain conditions (Zhang & He (2018)). We believe convergence analysis of MD for NNs could constitute to a completely new theoretical paper.

## 4 RELATED WORK

In this work we consider parameter quantization, which is usually formulated as a constrained problem and optimized using a modified projected gradient descent algorithm, where the methods (Courbariaux et al. (2015); Carreira-Perpiñán & Idelbayev (2017); Yin et al. (2018); Bai et al. (2019); Ajanthan et al. (2019)) mainly differ in the constraint set, the projection used, and how backpropagation through the projection is performed. Among them, STE based gradient descent is the most popular method as it enables backpropagation through nondifferentiable projections and it has shown to be highly effective in practice (Courbariaux et al. (2015)). In fact, the success of this approach lead to various extensions by including additional layerwise scalars (Rastegari et al. (2016)), relaxing the solution space (Yin et al. (2018)), and even to quantizing activations (Hubara et al. (2017)), and/or gradients (Zhou et al. (2016)). Moreover, there are methods focusing on loss aware quantization (Hou et al. (2017)), quantization for specialized hardware (Esser et al. (2015)), and quantization based on the variational approach (Achterhold et al. (2018); Louizos et al. (2017; 2019)). We have only provided a brief summary of relevant methods and for a comprehensive survey we refer the reader to (Guo (2018)).

## 5 EXPERIMENTS

Due to the popularity of binary neural networks (Courbariaux et al. (2015); Rastegari et al. (2016)), we mainly consider binary quantization and set the quantization levels as $\mathcal{Q} = \{-1, 1\}$. We would

like to point out that we quantize all learnable parameters, meaning our approach results in 32 times less memory compared to the floating-point counterparts.

We evaluate two MD variants corresponding to $\tanh$ and $\text{softmax}$ projections, on CIFAR-10, CIFAR-100 and TinyImageNet[2] datasets with VGG-16 and ResNet-18 architectures. We also evaluate the numerically stable versions of our MD variants (denoted with suffix "-S") performed by storing auxiliary parameters during training as explained in Eq. (22). The results are compared against parameter quantization methods, namely BinaryConnect (BC) (Courbariaux et al. (2015)), ProxQuant (PQ) (Bai et al. (2019)) and Proximal Mean-Field (PMF) (Ajanthan et al. (2019)). In addition, for completeness, we also compare against a standard PGD variant corresponding to the $\tanh$ projection (denoted as GD-tanh), *i.e.*, minimizing $f(\tanh(\tilde{\mathbf{x}}))$ using gradient descent. The only difference of this to our MD-tanh-S is that, in Eq. (22), the Jacobian of $\tanh$ is directly used in the updates. Note that, numerous techniques have emerged with BC as the workhorse algorithm by relaxing constraints such as the layer-wise scalars (Rastegari et al. (2016)), and similar extensions are straightforward even in our case. Briefly, our results indicate that the binary networks obtained by the MD variants yield accuracies very close to the floating-point counterparts while outperforming the baselines.

For all the experiments, standard multi-class cross-entropy loss is used. We crossvalidate the hyperparameters such as learning rate, learning rate scale, rate of increase of annealing hyperparameter $\beta$, and their respective schedules for all tested methods including the baselines. This extensive crossvalidation improved the accuracies of previous methods by a large margin, e.g*., up to $3\%$ improvement for* PMF. We provide the hyperparameter tuning search space and the final hyperparameters in Appendix C. Our algorithm is implemented in PyTorch (Paszke et al. (2017)) and the experiments are performed on NVIDIA Tesla-P100 GPUs. Our code will be released upon publication.

| | Algorithm | Space | CIFAR-10 | | CIFAR-100 | | TinyImageNet |
| | | | VGG-16 | ResNet-18 | VGG-16 | ResNet-18 | ResNet-18 |
|---|---|---|---|---|---|---|---|
| | REF (float) | **w** | 93.33 | 94.84 | 71.50 | 76.31 | 58.35 |
| | BC | **w** | 89.04 | 91.64 | 59.13 | 72.14 | 49.65 |
| | PQ | **w** | 85.41 | 90.76 | 39.61 | 65.13 | 44.32 |
| | PQ* | **w** | 90.11 | 92.32 | 55.10 | 68.35 | 49.97 |
| | PMF | **u** | 90.51 | 92.73 | 61.52 | 71.85 | 51.00 |
| | PMF* | **u** | 91.40 | 93.24 | **64.71** | 71.56 | 51.52 |
| | GD-tanh | **w** | 91.47 | 93.27 | 60.67 | 71.46 | 51.43 |
| Ours | MD-softmax | **u** | 90.47 | 91.28 | 56.25 | 68.49 | 46.52 |
| | MD-tanh | **w** | **91.64** | 92.97 | 61.31 | 72.13 | **54.62** |
| | MD-softmax-S | **u** | 91.30 | **93.28** | 63.97 | **72.18** | 51.81 |
| | MD-tanh-S | **w** | 91.53 | 93.18 | 61.69 | **72.18** | 52.32 |

Table 1: *Classification accuracies on the test set for different methods for binary quantization.* PQ* *denotes performance with biases, fully-connected layers, and shortcut layers in floating-point (original* PQ *setup) whereas* PQ *represents full quantization.* PMF* *denotes the performance of* PMF *after crossvalidation similar to our* MD*-variants and the original results from the paper are denoted as* PMF*. Note our* MD *variants obtained accuracies virtually the same as the best performing method and it outperformed the best method by a large margin in much harder TinyImageNet dataset.*

## 5.1 RESULTS

The classification accuracies of binary networks obtained by both variants of our algorithm, namely, MD-tanh and MD-softmax, their numerically stable versions (suffix "-S") and the baselines BC, PQ, PMF, GD-tanh and the floating point Reference Network (REF) are reported in Table 1. Both the numerically stable MD variants consistently produce better or on par results compared to other binarization methods while narrowing the performance gap between binary networks and floating point counterparts to a large extent, on multiple datasets.

Our stable MD-variant perform slightly better than MD-softmax, whereas for $\tanh$, MD updates either perform on par or sometimes even better than numerically stable version of MD-tanh. We

---

[2]`https://tiny-imagenet.herokuapp.com/`

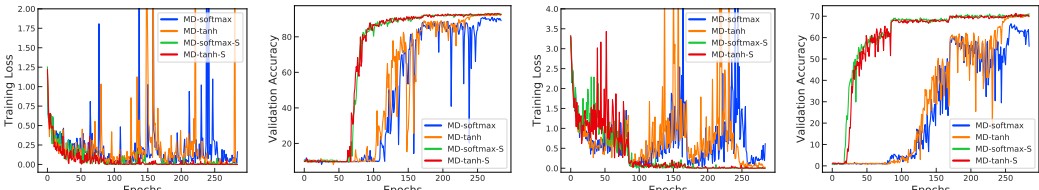

Figure 3: *Training curves for binarization for CIFAR-10 (first two) and CIFAR-100 (last two) with ResNet-18. Compared to original* MD *variants, stable* MD *variants are less noisy and after the initial exploration phase (up to* 60 *in CIFAR-10 and* 25 *epochs CIFAR-100), the validation accuracies rise sharply and show gradual improvement afterwards.*

believe, the main reason for this empirical variation in results for our MD-variants is due to numerical instability caused by the floating-point arithmetic of logarithm and exponential functions in Eq. (15) and Eq. (20). Furthermore, even though our two MD-variants, namely MD-softmax and MD-tanh optimize in different spaces, their performance is similar in most cases. This may be explained by the fact that both algorithms belong to the same family where a "soft" projection to the constraint set (in fact the constraints sets are equivalent in this case, refer Sec. 2.2.2) is used and an annealing hyperparameter is used to gradually enforce a quantized solution.

Note, PQ (Bai et al. (2019)) does not quantizee the fully connected layers, biases and shortcut layers. For fair comparison, we crossvalidate PQ with all layers binarized and original PQ settings, and report the results denoted as PQ and PQ* respectively in Table 1. Our MD-variants outperform PQ consistently on multiple datasets in equivalent experimental settings. This clearly shows that entropic or tanh-based regularization with our annealing scheme is superior to a simple "W" shaped regularizer and emphasizes that MD is a suitable framework for quantization.

Furthermore, the superior performance of MD-tanh against GD-tanh and on par or better performance of MD-softmax against PMF for binary quantization empirically validates that MD is useful even in a nonconvex stochastic setting. This hypothesis along with our numerically stable form of MD can be particularly useful to explore other projections which are useful for quantization and/or network compression in general.

The training curves for our MD variants for CIFAR-10 and CIFAR-100 datasets with ResNet-18 are shown in Fig. 3. The original MD variants show unstable behaviour during training which is attributed to the fact that it involves logarithms and exponentials in the update rules. In addition, we believe, the additional annealing hyperparameter also contributes to this instability. Regardless, by storing auxiliary variables, the MD updates are demonstrated to be quite stable. This clear distinction between MD variants emphasizes the significance of practical considerations while implementing MD especially in NN optimization. For more experiments such as training curves comparison to other methods and ternary quantization results please refer to the Appendix C.

## 6 DISCUSSION

In this work, we have introduced an MD framework for NN quantization by deriving mirror maps corresponding to various projections useful for quantization. In addition, we have discussed a numerically stable implementation of MD by storing an additional set of auxiliary variables and showed that this update is strikingly analogous to the popular STE based gradient method. The superior performance of our MD formulation even with simple projections such as tanh and softmax is encouraging and we believe, MD would be a suitable framework for not just NN quantization but for network compression in general. Finally, some theoretical aspects such as the use of time-varying mirror maps and the combination of MD and a stochastic optimizer such as Adam are left unattended in this paper, which we intend to analyze in a future work.

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

# Appendices

Here, we first provide the proof of the theorem and the technical derivations. Later we give additional experiments and the details of our experimental setting.

## A    DERIVING MIRROR MAPS FROM PROJECTIONS

**Theorem A.1.** *Let $\mathcal{X}$ be a compact convex set and $P : \mathbb{R} \to \mathcal{C}$ be an invertible function where $\mathcal{C} \subset \mathbb{R}$ is a convex open set such that $\mathcal{X} = \bar{\mathcal{C}}$ ($\bar{\mathcal{C}}$ denotes the closure of $\mathcal{C}$). Now, if*

*1. $P$ is strictly monotonically increasing.*
*2. $\lim_{x \to \partial \mathcal{C}} \|P^{-1}(x)\| = \infty$ ($\partial \mathcal{C}$ denotes the boundary of $\mathcal{C}$).*

*Then, $\Phi(x) = \int_{x_0}^{x} P^{-1}(y)dy$ is a valid mirror map.*

*Proof.* From the fundamental theorem of calculus, the gradient of $\Phi(x)$ satisfies, $\nabla \Phi(x) = P^{-1}(x)$. Since $P$ is strictly monotonically increasing and invertible, $P^{-1}$ is strictly monotonically increasing. Therefore, $\Phi(x)$ is strictly convex and differentiable. Now, from the definition of projection and since it is invertible (*i.e.*, $P^{-1}$ is *one-to-one* and *onto*), $\nabla \Phi(\mathcal{C}) = P^{-1}(\mathcal{C}) = \mathbb{R}$. Therefore, together with condition (2), we can conclude that $\Phi(x) = \int_{x_0}^{x} P^{-1}(y)dy$ is a valid mirror map (refer Definition 2.2 in the main paper). For the multi-dimensional case, we need an additional condition that the vector field $P^{-1}(\mathbf{x})$ is conservative. Then by the gradient theorem (gra), there exists a mirror map $\Phi(\mathbf{x}) = \int_{\mathbf{x}_0}^{\mathbf{x}} P^{-1}(\mathbf{y})d\mathbf{y}$ for some arbitrary base point $\mathbf{x}_0$.

$\square$

## B  MD UPDATE DERIVATION FOR THE tanh PROJECTION

We now derive the MD update corresponding to the tanh projection below. From Theorem A.1, the mirror map for the tanh projection can be written as:

$$\Phi(w) = \int P^{-1}(w)dw = \frac{1}{2\beta}\big[(1+w)\log(1+w) + (1-w)\log(1-w)\big] . \tag{24}$$

Correspondingly, the Bregman divergence can be written as:

$$D_\Phi(w,v) = \Phi(w) - \Phi(v) - \Phi'(v)(w-v) , \quad \text{where } \Phi'(v) = \frac{1}{2\beta}\log\frac{1+v}{1-v} , \tag{25}$$

$$= \frac{1}{2\beta}\left[ w\log\frac{(1+w)(1-v)}{(1-w)(1+v)} + \log(1-w)(1+w) - \log(1-v)(1-v) \right] .$$

Now, consider the proximal form of MD update

$$w^{k+1} = \underset{\mathbf{x}\in(-1,1)}{\text{argmin}} \langle \eta\, g^k, w\rangle + D_\Phi(w, w^k) . \tag{26}$$

The idea is to find $w$ such that the KKT conditions are satisfied. To this end, let us first write the Lagrangian of Eq. (26) by introducing dual variables $y$ and $z$ corresponding to the constraints $w > -1$ and $w < 1$, respectively:

$$F(w,x,y) = \eta g^k w + \frac{1}{2\beta}\left[ w\log\frac{(1+w)(1-w^k)}{(1-w)(1+w^k)} + \log(1-w)(1+w) - \log(1-w^k)(1-w^k) \right] \tag{27}$$

$$+ y(-w-1) + z(w-1) .$$

Now, setting the derivatives with respect to $w$ to zero:

$$\frac{\partial F}{\partial w} = \eta g^k + \frac{1}{2\beta}\log\frac{(1+w)(1-w^k)}{(1-w)(1+w^k)} - y + z = 0 . \tag{28}$$

From complementary slackness conditions,

$$y(-w-1) = 0 , \qquad \text{since} \quad w > -1 \Rightarrow y = 0 , \tag{29}$$
$$z(w-1) = 0 , \qquad \text{since} \quad w < 1 \Rightarrow z = 0 .$$

Therefore, Eq. (28) now simplifies to:

$$\frac{\partial F}{\partial w} = \eta g^k + \frac{1}{2\beta}\log\frac{(1+w)(1-w^k)}{(1-w)(1+w^k)} = 0 , \tag{30}$$

$$\log\frac{(1+w)(1-w^k)}{(1-w)(1+w^k)} = \exp(-2\beta\eta g^k) ,$$

$$\frac{1+w}{1-w} = \frac{1+w^k}{1-w^k}\exp(-2\beta\eta g^k) ,$$

$$w = \frac{\frac{1+w^k}{1-w^k}\exp(-2\beta\eta g^k) - 1}{\frac{1+w^k}{1-w^k}\exp(-2\beta\eta g^k) + 1} .$$

The pseudocodes of original (MD-tanh) and numerically stable versions (MD-tanh-S) for tanh are presented in Algorithms 1 and 2 respectively.

## C  ADDITIONAL EXPERIMENTS

We first give training curves of all compared methods, and provide ternary quantization results as a proof of concept. Later, we provide experimental details.

**Convergence Analysis.**  The training curves for CIFAR-10 and CIFAR-100 datasets with ResNet-18 are shown in Fig. 4. Notice, after the initial exploration phase (due to low $\beta$) the validation accuracies of our MD-tanh-S increase sharply while this steep rise is not observed in regularization methods such as PQ. The training behaviour for both our stable MD-variants (softmax and tanh) is quite similar.

---

**Algorithm 1** MD-tanh

---

**Require:** $K, b, \{\eta^k\}, \rho > 1, \mathcal{D}, L$
**Ensure:** $\mathbf{w}^* \in \mathcal{Q}^m$
 1: $\mathbf{w}^0 \in \mathbb{R}^m, \quad \beta \leftarrow 1$ ▷ Initialization
 2: $\mathbf{w}^0 \leftarrow \tanh(\beta \mathbf{w}^0)$ ▷ Projection
 3: **for** $k \leftarrow 0, \ldots, K$ **do**
 4: $\quad \mathcal{D}^b = \{(\mathbf{x}_i, \mathbf{y}_i)\}_{i=1}^b \sim \mathcal{D}$ ▷ Sample a mini-batch
 5: $\quad \mathbf{g}^k \leftarrow \nabla_\mathbf{w} L(\mathbf{w}; \mathcal{D}^b)\big|_{\mathbf{w}=\mathbf{w}^k}$ ▷ Gradient w.r.t. $\mathbf{w}$ at $\mathbf{w}^k$ (Adam based gradients)
 6: $\quad$ **for** $j \leftarrow 1, \ldots, m$ **do**
 7: $\quad\quad w_j^{k+1} \leftarrow \dfrac{\frac{1+w_j^k}{1-w_j^k} \exp(-2\beta\eta^k g_j^k) - 1}{\frac{1+w_j^k}{1-w_j^k} \exp(-2\beta\eta^k g_j^k) + 1}$ ▷ MD update
 8: $\quad$ **end for**
 9: $\quad \beta \leftarrow \rho\beta$ ▷ Increase $\beta$
10: **end for**
11: $\mathbf{w}^* \leftarrow \text{sign}(\tilde{\mathbf{w}}^K)$ ▷ Quantization

---

**Algorithm 2** MD-tanh-S

---

**Require:** $K, b, \{\eta^k\}, \rho > 1, \mathcal{D}, L$
**Ensure:** $\mathbf{w}^* \in \mathcal{Q}^m$
 1: $\tilde{\mathbf{w}}^0 \in \mathbb{R}^m, \quad \beta \leftarrow 1$ ▷ Initialization
 2: **for** $k \leftarrow 0, \ldots, K$ **do**
 3: $\quad \mathbf{w}^k \leftarrow \tanh(\beta \tilde{\mathbf{w}}^k)$ ▷ Projection
 4: $\quad \mathcal{D}^b = \{(\mathbf{x}_i, \mathbf{y}_i)\}_{i=1}^b \sim \mathcal{D}$ ▷ Sample a mini-batch
 5: $\quad \mathbf{g}^k \leftarrow \nabla_\mathbf{w} L(\mathbf{w}; \mathcal{D}^b)\big|_{\mathbf{w}=\mathbf{w}^k}$ ▷ Gradient w.r.t. $\mathbf{w}$ at $\mathbf{w}^k$ (Adam based gradients)
 6: $\quad \tilde{\mathbf{w}}^{k+1} \leftarrow \tilde{\mathbf{w}}^k - \eta^k \mathbf{g}^k$ ▷ Gradient descent on $\tilde{\mathbf{w}}$
 7: $\quad \beta \leftarrow \rho\beta$ ▷ Increase $\beta$
 8: **end for**
 9: $\mathbf{w}^* \leftarrow \text{sign}(\tilde{\mathbf{w}}^K)$ ▷ Quantization

---

**Ternary Quantization.** As a proof of concept for our shifted $\tanh$ projection (refer Example 3.3), we also show results for ternary quantization with quantization levels $\mathcal{Q} = \{-1, 0, 1\}$ in Table 2. Note that the performance improvement of our ternary networks compared to their respective binary networks is marginal as only 0 is included as the 3rd quantization level. In contrast to us, the baseline method PQ (Bai et al. (2019)) optimizes for the quantization levels (differently for each layer) as well in an alternating optimization regime rather than fixing it to $\mathcal{Q} = \{-1, 0, 1\}$. Also, PQ does ternarize the first convolution layer, fully-connected layers and the shortcut layers. We cross-validate hyperparameters for both the original PQ setup and the equivalent setting of our MD-variants where we optimize all the weights and denote them as PQ* and PQ respectively.

Our MD-tanh variant performs on par or sometimes even better in comparison to $\tanh$ projection results where gradient is calculated through the projection instead of performing MD. This again empirically validates the hypothesis that MD yields in good approximation for the task of network quantization. The better performance of PQ in their original quantization setup, compared to our approach in CIFAR-10 can be accounted to their non-quantized layers and different quantization levels. We believe, similar explorations are possible with our MD framework as well.

**Experimental Details.** As mentioned in the main paper the experimental protocol is similar to (Ajanthan et al. (2019)). To this end, the details of the datasets and their corresponding experiment setups are given in Table 3. For CIFAR-10/100 and TinyImageNet, VGG-16 (Simonyan & Zisserman (2015)) and ResNet-18 (He et al. (2016)) architectures adapted for CIFAR dataset are used. In particular, for CIFAR experiments, similar to (Lee et al. (2019)), the size of the fully-connected (FC) layers of VGG-16 is set to 512 and no dropout layers are employed. For TinyImageNet, the stride of the first convolutional layer of ResNet-18 is set to 2 to handle the image size (Huang et al. (2017)). In all the models, batch normalization (Ioffe & Szegedy (2015)) (with no learnable parameters) and

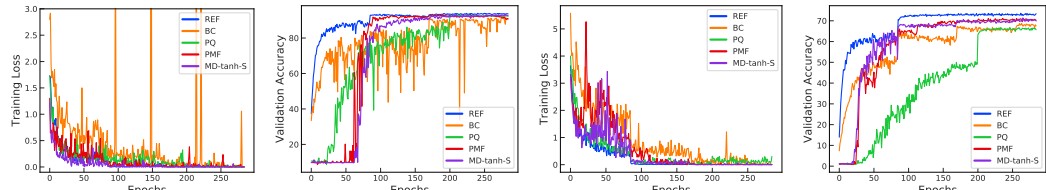

Figure 4: *Training curves for binarization for CIFAR-10 (first two columns) and CIFAR-100 (last two columns) with ResNet-18. Compared to* BC, *our* MD-tanh-S *and* PMF *are less noisy and after the initial exploration phase (up to* 60 *in CIFAR-10 and* 25 *epochs CIFAR-100), the validation accuracies rise sharply and closely resembles the floating point network afterwards. This steep increase is not observed in regularization methods such as* PQ.

| | Algorithm | Space | CIFAR-10 | | CIFAR-100 | | TinyImageNet |
| | | | VGG-16 | ResNet-18 | VGG-16 | ResNet-18 | ResNet-18 |
|---|---|---|---|---|---|---|---|
| | REF (float) | **w** | 93.33 | 94.84 | 71.50 | 76.31 | 58.35 |
| | PQ | **w** | 83.32 | 90.50 | 32.16 | 59.18 | 41.46 |
| | PQ* | **w** | 92.20 | 93.85 | 57.64 | 70.98 | 45.72 |
| | GD-tanh | **w** | 91.21 | 93.20 | 53.88 | 69.48 | 50.65 |
| Ours | MD-softmax-S | **u** | 91.69 | 93.30 | 65.11 | 72.01 | 52.21 |
| | MD-tanh-S | **w** | 91.70 | 93.42 | 66.15 | 71.29 | 52.69 |

Table 2: *Classification accuracies on the test set for ternary quantization.* PQ* *denotes performance with fully-connected layers, first convolution layer and shortcut layers in floating point whereas* PQ *represent results with all layers quantized. Also,* PQ* *optimize for the quantization levels as well (different for each layer), in contrast we fix it to* $\mathcal{Q} = \{-1, 0, 1\}$. GD-tanh *denotes results without using* STE *and actually calculating the gradient through the projection.*

ReLU nonlinearity are used. Only for the floating point networks (*i.e.*, REF), we keep the learnable parameters for batch norm. Standard data augmentation (*i.e.*, random crop and horizontal flip) is used.

For both of our MD variants, hyperparameters such as the learning rate, learning rate scale, annealing hyperparameter $\beta$ and its schedule are crossvalidated from the range reported in Table 4 and the chosen parameters are given in the Tables 5 and 6. To generate the plots, we used the publicly available codes of BC[3], PQ[4] and PMF[5].

All methods are trained from a random initialization and the model with the best validation accuracy is chosen for each method. Note that, in MD, even though we use an increasing schedule for $\beta$ to enforce a discrete solution, the chosen network may not be fully-quantized (as the best model could be obtained in an early stage of training). Therefore, simple argmax rounding is applied to ensure that the network is fully-quantized.

---

[3] https://github.com/itayhubara/BinaryNet.pytorch
[4] https://github.com/allenbai01/ProxQuant
[5] https://github.com/tajanthan/pmf

| Dataset | Image | # class | Train / Val. | $b$ | $K$ |
|---|---|---|---|---|---|
| MNIST | $28 \times 28$ | 10 | 50k / 10k | 100 | 20k |
| CIFAR-10 | $32 \times 32$ | 10 | 45k / 5k | 128 | 100k |
| CIFAR-100 | $32 \times 32$ | 100 | 45k / 5k | 128 | 100k |
| TinyImageNet | $64 \times 64$ | 200 | 100k / 10k | 128 | 100k |

| Hyperparameters | Fine-tuning grid |
|---|---|
| learning_rate | $[0.1, 0.01, 0.001, 0.0001]$ |
| lr_scale | $[0.1, 0.2, 0.3, 0.5]$ |
| beta_scale | $[1.01, 1.02, 1.05, 1.1, 1.2]$ |
| beta_scale_interval | $[100, 200, 500, 1000, 2000]$ |

Table 3: *Experiment setup. Here, $b$ is the batch size and $K$ is the total number of iterations for all the methods.*

Table 4: *The hyperparameter search space for all the experiments. Chosen parameters are given in Tables 5 and 6.*

| | CIFAR-10 with ResNet-18 | | | | | | | |
|---|---|---|---|---|---|---|---|---|
| | MD-softmax | MD-tanh | MD-softmax-S | MD-tanh-S | PMF* | GD-tanh | BC | PQ |
| learning_rate | 0.001 | 0.001 | 0.001 | 0.001 | 0.001 | 0.001 | 0.001 | 0.01 |
| lr_scale | 0.2 | 0.3 | 0.3 | 0.3 | 0.3 | 0.3 | 0.3 | 0.5 |
| beta_scale | 1.02 | 1.01 | 1.02 | 1.02 | 1.1 | 1.1 | - | 0.0001 |
| beta_scale_interval | 200 | 100 | 200 | 200 | 1000 | 1000 | - | - |

| | CIFAR-100 with ResNet-18 | | | | | | | |
|---|---|---|---|---|---|---|---|---|
| | MD-softmax | MD-tanh | MD-softmax-S | MD-tanh-S | PMF* | GD-tanh | BC | PQ |
| learning_rate | 0.001 | 0.001 | 0.001 | 0.001 | 0.001 | 0.001 | 0.001 | 0.1 |
| lr_scale | 0.2 | 0.3 | 0.2 | 0.2 | 0.3 | 0.5 | 0.2 | - |
| beta_scale | 1.05 | 1.05 | 1.1 | 1.2 | 1.01 | 1.01 | - | 0.001 |
| beta_scale_interval | 500 | 500 | 200 | 500 | 100 | 100 | - | - |

| | CIFAR-10 with VGG-16 | | | | | | | |
|---|---|---|---|---|---|---|---|---|
| | MD-softmax | MD-tanh | MD-softmax-S | MD-tanh-S | PMF* | GD-tanh | BC | PQ |
| learning_rate | 0.01 | 0.001 | 0.001 | 0.001 | 0.001 | 0.001 | 0.0001 | 0.01 |
| lr_scale | 0.2 | 0.3 | 0.3 | 0.2 | 0.5 | 0.3 | 0.3 | 0.5 |
| beta_scale | 1.05 | 1.1 | 1.2 | 1.2 | 1.05 | 1.1 | - | 0.0001 |
| beta_scale_interval | 500 | 1000 | 2000 | 2000 | 500 | 1000 | - | - |

| | CIFAR-100 with VGG-16 | | | | | | | |
|---|---|---|---|---|---|---|---|---|
| | MD-softmax | MD-tanh | MD-softmax-S | MD-tanh-S | PMF* | GD-tanh | BC | PQ |
| learning_rate | 0.001 | 0.001 | 0.0001 | 0.001 | 0.0001 | 0.001 | 0.0001 | 0.01 |
| lr_scale | 0.3 | 0.3 | 0.2 | 0.5 | 0.5 | 0.5 | 0.2 | 0.5 |
| beta_scale | 1.01 | 1.05 | 1.2 | 1.05 | 1.02 | 1.1 | - | 0.0001 |
| beta_scale_interval | 100 | 500 | 500 | 500 | 200 | 1000 | - | - |

| | TinyImageNet with ResNet-18 | | | | | | | |
|---|---|---|---|---|---|---|---|---|
| | MD-softmax | MD-tanh | MD-softmax-S | MD-tanh-S | PMF* | GD-tanh | BC | PQ |
| learning_rate | 0.001 | 0.001 | 0.001 | 0.001 | 0.001 | 0.001 | 0.001 | 0.01 |
| lr_scale | 0.2 | 0.5 | 0.1 | 0.1 | 0.5 | 0.5 | 0.5 | - |
| beta_scale (ours) | 1.02 | 1.2 | 1.02 | 1.2 | 1.01 | 1.01 | - | 0.0001 |
| beta_scale_interval | 200 | 2000 | 100 | 500 | 100 | 100 | - | - |

Table 5: *Hyperparameter settings used for the binary quantization experiments. Here, the learning rate is multiplied by* lr_scale *after every 30k iterations and annealing hyperparameter $(\beta)$ is multiplied by* beta_scale *after every* beta_scale_interval *iterations. We use Adam optimizer with zero weight decay. For* PQ, beta_scale *denotes regularization rate.*

| | CIFAR-10 with ResNet-18 | | | | | |
| --- | --- | --- | --- | --- | --- | --- |
| | REF (float) | MD-softmax-S | MD-tanh-S | GD-tanh | PQ | PQ* |
| learning_rate | 0.1 | 0.001 | 0.01 | 0.01 | 0.01 | 0.01 |
| lr_scale | 0.3 | 0.3 | 0.2 | 0.5 | 0.3 | - |
| beta_scale (ours) | - | 1.05 | 1.2 | 1.02 | 0.0001 | 0.0001 |
| beta_scale_interval | - | 500 | 1000 | 500 | - | - |
| weight_decay | 0.0001 | 0 | 0 | 0 | 0 | 0.0001 |
| | CIFAR-100 with ResNet-18 | | | | | |
| | REF (float) | MD-softmax-S | MD-tanh-S | GD-tanh | PQ | PQ* |
| learning_rate | 0.1 | 0.001 | 0.001 | 0.01 | 0.01 | 0.001 |
| lr_scale | 0.1 | 0.1 | 0.5 | 0.5 | 0.2 | - |
| beta_scale (ours) | - | 1.1 | 1.1 | 1.02 | 0.0001 | 0.0001 |
| beta_scale_interval | - | 100 | 500 | 1000 | - | - |
| weight_decay | 0.0001 | 0 | 0 | 0 | 0 | 0.0001 |
| | CIFAR-10 with VGG-16 | | | | | |
| | REF (float) | MD-softmax-S | MD-tanh-S | GD-tanh | PQ | PQ* |
| learning_rate | 0.1 | 0.001 | 0.01 | 0.01 | 0.01 | 0.1 |
| lr_scale | 0.2 | 0.3 | 0.3 | 0.3 | - | - |
| beta_scale (ours) | - | 1.05 | 1.1 | 1.01 | 1e-07 | 0.0001 |
| beta_scale_interval | - | 500 | 1000 | 500 | - | - |
| weight_decay | 0.0001 | 0 | 0 | 0 | 0 | 0.0001 |
| | CIFAR-100 with VGG-16 | | | | | |
| | REF (float) | MD-softmax-S | MD-tanh-S | GD-tanh | PQ | PQ* |
| learning_rate | 0.1 | 0.0001 | 0.001 | 0.01 | 0.01 | 0.0001 |
| lr_scale | 0.2 | 0.3 | 0.5 | 0.2 | - | - |
| beta_scale (ours) | - | 1.05 | 1.1 | 1.05 | 0.0001 | 0.0001 |
| beta_scale_interval | - | 100 | 500 | 2000 | - | - |
| weight_decay | 0.0001 | 0 | 0 | 0 | 0 | 0.0001 |
| | TinyImageNet with ResNet-18 | | | | | |
| | REF (float) | MD-softmax-S | MD-tanh-S | GD-tanh | PQ | PQ* |
| learning_rate | 0.1 | 0.001 | 0.01 | 0.01 | 0.01 | 0.01 |
| lr_scale | 0.1 | 0.1 | 0.1 | 0.5 | - | - |
| beta_scale (ours) | - | 1.2 | 1.2 | 1.05 | 0.01 | 0.0001 |
| beta_scale_interval | - | 500 | 2000 | 2000 | - | - |
| weight_decay | 0.0001 | 0 | 0 | 0 | 0 | 0.0001 |

Table 6: *Hyperparameter settings used for the ternary quantization experiments. Here, the learning rate is multiplied by* lr_scale *after every 30k iterations and annealing hyperparameter* $(\beta)$ *is multiplied by* beta_scale *after every* beta_scale_interval *iterations. We use Adam optimizer except for* REF *for which* SGD *with momentum* 0.9 *is used. For* PQ*,* beta_scale *denotes regularization rate.*

