# OpenReview forum: "Mirror Descent View For Neural Network Quantization"
_ICLR.cc/2020/Conference — Reject_

### Official Review · AnonReviewer3 · 2019-10-22
**Official Blind Review #3**

**Rating:** 3

**Review:**

This paper proposes a neural network (NN) quantization based on Mirror Descent (MD) framework. The core of the proposal is the construction of the mirror map from the unconstrained auxiliary variables to the quantized space. Building on that core, the authors derive some mapping functions from the corresponding projection, i.e. tanh, softmax and shifted tanh. The experimental result on benchmark datasets (CIFAR & TinyImageNet) and basic architectures (VGG & ResNet-18) showed that the proposed method is suitable for quantization. The proposed method is a natural extension of ProxQuant, which adopted the proximal gradient descent to quantize NN (a.k.a $\ell_2$ norm in MD). Different projections in NN quantization lead to different Bregman divergences in MD.

However, the authors do not analyze the convergence of the MD with nonconvex objective function in NN quantization neither how to choose the projection for mirror mapping construction. Moreover, it is better to discuss with [Bai et al, 2019] to clarify the novelty of the proposed method. So I concern about the novelty and the theoretical contributions

Yu Bai, Yu-Xiang Wang, Edo Liberty.
ProxQuant: Quantized Neural Networks via Proximal Operators. ICLR 2019.

**Experience Assessment:**

I have published one or two papers in this area.

**Review Assessment: Checking Correctness Of Derivations And Theory:**

I carefully checked the derivations and theory.

**Review Assessment: Checking Correctness Of Experiments:**

I assessed the sensibility of the experiments.

**Review Assessment: Thoroughness In Paper Reading:**

I read the paper thoroughly.

---

> ### Author Response · Authors · 2019-11-08
> **Summary of contributions and response to other comments {Response to R3 [2/2]}**
>
> Below, we first summarize our contributions and then address the comments regarding convergence analysis and choice of projection.
>
> # Main contributions of our MD method
> - We would like to clarify that the main focus of the paper is to show that MD is a suitable framework for NN quantization and introduce a practical MD algorithm for NN quantization. To this end, our main contributions are summarized below:
> - We introduce the first practical MD based algorithm for NN quantization with time-varying mirror maps and demonstrate superior empirical performance against directly comparable baselines.
> - As MD updates are prone to numerical instability, we introduce a numerically stable version of MD and show that the popular STE method is an implementation method for MD under certain conditions on the projection.
>
> # Convergence of MD in the nonconvex setting
> - As mentioned in our submission, convergence analysis of MD in the nonconvex setting is an active research area [3,4] and MD has been recently shown to converge in the nonconvex stochastic setting under certain conditions [5]. This, together with empirical convergence plots (in Fig. 2) justifies the use of MD for NN quantization. We have appropriately cited [5] in the revised version and we believe convergence analysis of MD for NNs could be a completely new theoretical paper in itself.
>
> # Choice of projection
> - As stated in Theorem 3.1, if the projection is invertible and monotonically increasing, a valid mirror map exists and the corresponding MD algorithm can be derived. Moreover, to ensure fully-quantized networks we require the projections to be parameterized by an annealing hyperparameter $\beta$ (Example 3.1). Nevertheless, choosing a projection that is guaranteed to yield improved quantization performance is an open problem.
>
> [3] Zhou, Zhengyuan, Panayotis Mertikopoulos, Nicholas Bambos, Stephen Boyd, and Peter Glynn. "Mirror descent in non-convex stochastic programming." CoRR (2017).
> [4] Zhou, Zhengyuan, Panayotis Mertikopoulos, Nicholas Bambos, Stephen Boyd, and Peter W. Glynn. "Stochastic mirror descent in variationally coherent optimization problems." NeurIPS (2017).
> [5] Zhang, Siqi, and Niao He. "On the convergence rate of stochastic mirror descent for nonsmooth nonconvex optimization." CoRR (2018).

---

> ### Author Response · Authors · 2019-11-08
> **PQ is not based on MD and our MD method significantly outperforms PQ {Response to R3 [1/2]}**
>
> Thank you for the feedback and we appreciate that the reviewer finds that our MD method is suitable for NN quantization.
>
> In this reply, we clarify the novelty and significance of our MD method compared to ProxQuant (PQ) [1]. Meanwhile, responses to other comments will be provided in a subsequent reply.
>
> # Summary
> - Our main contribution of the paper is to show that MD is a suitable framework for NN quantization and introduce a numerically stable MD algorithm for NN quantization with superior performance compared to directly comparable baselines.
> - In this regard, we find the statement that our MD method is a “natural extension of PQ” (ie, proximal gradient method or in general gradient descent where the $L_2$ norm is used) to be misleading and the differences are as follows.
>
> # MD vs PQ
> - The main and important difference between our MD method and PQ is that MD allows gradient descent to be performed on a more general non-Euclidean space (refer to Sec. 2) whereas PQ does not. To see this, we first give the update equations of PQ and MD below:
> - PQ: $\tilde{x}^{k+1} \gets x^k - \eta g^k$ where $x^k = \text{prox}(\tilde{x}^k)$ and $g^k = \nabla f(x)|_{x = x^k}$. Here, $x^k, \tilde{x}^k \in R$. (refer to Alg. 1 in [1])
> - MD: $\tilde{x}^{k+1} \gets \tilde{x}^k - \eta g^k$ where $x^k = P(\tilde{x}^k)$ and $g^k = \nabla f(x)|_{x = x^k}$. Here, $x^k \in B$ and $\tilde{x}^k \in B^*$, where $B^*$ is the dual space of $B$. (refer to Eq. 22 in the paper)
> - Notice that, PQ assumes the point $x^k$ and gradient $g^k$ are in the same space. Then only the formula $x^k - \eta g^k$ is valid. This would only be true for the Euclidean space [2]. However, MD allows gradient descent to be performed on a more general non-Euclidean space by first mapping a primal point $x^k\in B$ to a point $\tilde{x}^k \in B^*$ in the dual space via the mirror map. Such an ability is extremely beneficial in many problems (eg, simple constrained optimization) and it enabled theoretical and practical research on MD for the past three decades. Therefore, as mentioned in the paper (page 7) PQ is not based on MD.
> - Furthermore, it is clear from our experiments that MD significantly outperforms PQ (up to 20% in some cases when fully-quantized, refer to Table 1) demonstrating the importance of optimizing on a non-Euclidean space based on our MD framework.
> - Even though PQ hinted at the connection to the dual averaging version of MD and STE, it does not analyze the conditions on the projections under which corresponding valid mirror maps exist. This is important to show STE as a numerically stable implementation method for MD and such a link was previously lacking in the literature.
> - We have added this discussion in the revised version of the paper (page 7) to improve clarity.
>
> [1] Bai, Yu, Yu-Xiang Wang, and Edo Liberty. "Proxquant: Quantized neural networks via proximal operators." ICLR (2019).
> [2] Bubeck, Sébastien. "Convex optimization: Algorithms and complexity." Foundations and Trends® in Machine Learning (2015).

---

### Official Review · AnonReviewer2 · 2019-10-23
**Official Blind Review #2**

**Rating:** 6

**Review:**

This paper proposes a Mirror Descent (MD) framework for the quantization of neural networks, which, different with previous quantization methods, enables us to derive valid mirror maps and the respective MD updates. Moreover, the authors also provide a stable implementation of MD by storing an additional set of auxiliary dual variables. Experiments on CIFAR-10/100 and TinyImageNet with convolutional and residual architectures show the effective of the proposed model.

Overall, this paper is well-written and provide sufficient material, both theoretical and experimental evidence to support the proposed method. Although the novelty of this work is somehow limited, i.e. appling MD from convex optimization to NN quantization, the authors provides sufficient effort to explore how to success to adopted it the literature. Hence, I lean to make an accept suggestion at this point.

Concern: it would better to provide the code to validate the soundness of the model.

##post comments
The rebuttal addresses my concerns and I will not change my score. Thanks.

**Experience Assessment:**

I have read many papers in this area.

**Review Assessment: Checking Correctness Of Derivations And Theory:**

I assessed the sensibility of the derivations and theory.

**Review Assessment: Checking Correctness Of Experiments:**

I assessed the sensibility of the experiments.

**Review Assessment: Thoroughness In Paper Reading:**

I read the paper at least twice and used my best judgement in assessing the paper.

---

> ### Author Response · Authors · 2019-11-14
> **Thank you for the positive feedback**
>
> We appreciate that the reviewer finds that our paper has sufficient material on both theoretical and experimental aspects. Below we address the reviewer’s concerns.
>
> # Novelty
> - We appreciate that the reviewer acknowledges adopting MD for NN quantization has some challenges and our paper addresses them successfully (eg, time-varying mirror maps, deriving mirror maps from projections, numerically stable implementation using STE) and introduces the first practical MD based algorithm for NN quantization and demonstrate superior empirical performance against directly comparable baselines.
>
> # Code
> - We will provide the code upon publication and we have provided it for the reviewers and ACs in a separate confidential comment.

---

### Official Review · AnonReviewer1 · 2019-10-23
**Official Blind Review #1**

**Rating:** 8

**Review:**

A good paper that uses the Mirror Descent paradigm for learning quantized networks.
Though Mirror Descent is not their original idea, but using it in the context of learning quantized network is novel and interesting.
Empirically, they showed better results than existing method, with comparisons with reasonable baselines including using relaxed projected gradient descent.

Overall, I don’t have much concerns, but here are some more specific comment/questions (most relates to writing)

In the intro, it would be great to mention some past success on using MD, as opposed to just saying it’s well-known. Also you mention MD can be used for more than quantization, but compression in general, it’d be better to add that discussion, or remove this sentence.

In the beginning of Section 2.1, it'd be easier for the readers to make clear that the primal space corresponds to the quantized weights and the dual space corresponds to the unconstrained space in the rest of the paper.

At the top of page 3 you describe MD for the first time, but it’s unclear to me how y^0 is handled.

The end of section 3 and section 4 talk quite a bit about STE, maybe it'd be clear if the authors can provide a concise description.

As someone not super familiar with NN quantization, this work seems like a good contribution.  My only possible concerns would be somehow comparisons to existing methods are not comprehensive enough (if this will be pointed out by the other reviewers)



**Experience Assessment:**

I do not know much about this area.

**Review Assessment: Checking Correctness Of Derivations And Theory:**

I assessed the sensibility of the derivations and theory.

**Review Assessment: Checking Correctness Of Experiments:**

I assessed the sensibility of the experiments.

**Review Assessment: Thoroughness In Paper Reading:**

I read the paper thoroughly.

---

> ### Author Response · Authors · 2019-11-14
> **Thank you for the positive feedback**
>
> We appreciate that the reviewer finds that our method is novel and interesting. Below we address the reviewer’s concerns.
>
> # Writing suggestions
> - We agree with the reviewer’s suggestions and we have appropriately revised the paper.
> - Regarding $y^0$, we would like to clarify that the first iterate of $y$ is $y^1$ and it is obtained using Eq. 4 where $x^0$ is initialized as discussed in the paper.
>
> # Experiment setup
> - As discussed in the paper, not all NN quantization algorithms are directly comparable to each other due to variations in the experimental protocol and considered quantization levels [1]. Since it is impossible to evaluate on all different experimental setups, following the recent publications [2,3], we compare against directly comparable baselines and we consider the extreme case of fully-quantized networks (ie, all learnable parameters are quantized).
>
> [1] Guo, Yunhui. "A survey on methods and theories of quantized neural networks." CoRR (2018).
> [2] Bai, Yu, Yu-Xiang Wang, and Edo Liberty. "Proxquant: Quantized neural networks via proximal operators." ICLR (2019).
> [3] Ajanthan, Thalaiyasingam, Puneet K. Dokania, Richard Hartley, and Philip HS Torr. "Proximal Mean-field for Neural Network Quantization." ICCV (2019).

---

### Official Review · AnonReviewer4 · 2019-11-27
**Official Blind Review #4**

**Rating:** 3

**Review:**

The paper proposes to use the mirror descent algorithm for the binary network. The key point is Theorem 3.1, which enables the mirror map. The paper is easy to read and follow, and the main contributions are clearly stated.

However, I suggest a weak rejection of this paper. The reasons are

Q1. As Review #3, it is better for authors to provide more theoretical analysis, which better includes the nonconvex objective function and the effect of annealing.

Q2. It is not clear to me, why mirror descent is better than proximal gradient descent, i.e., proxQuant, in this application. The authors repeatedly claim "MD allows gradient descent to be performed on a more general non-Euclidean space". This cannot be told by Table 1, which is just overall performance. So, it is better to empirically show this point by an ablation study.

Q3. Since the technical contributions are not enough, I expect more experimental comparisons.
- Could the authors perform experiments on ImageNet?
- While VGG and ResNet are taken as a protocol for experimental comparison, it is better to do an extra comparison with STOA networks. VGG and ResNet are too old and easy to be compressed, compression these networks are of little practical values. EfficientNet [1], Mobilenets [2], and Shufflenet [3] can be good ones. The paper will be more convincing with these methods.

[1]. EfficientNet: Rethinking Model Scaling for Convolutional Neural Networks
[2]. Mobilenets: Efficient convolutional neural networks for mobile vision applications
[3]. Shufflenet: An extremely efficient convolutional neural network for mobile devices

**Experience Assessment:**

I have published one or two papers in this area.

**Review Assessment: Checking Correctness Of Derivations And Theory:**

I assessed the sensibility of the derivations and theory.

**Review Assessment: Checking Correctness Of Experiments:**

I assessed the sensibility of the experiments.

**Review Assessment: Thoroughness In Paper Reading:**

I read the paper thoroughly.

---

### Decision · Program_Chairs · 2019-12-19

**Decision:**

Reject

**Comment:**

The paper proposes to use the mirror descent algorithm for the binary network. It is easy to read. However, novelty over ProxQuant is somehow limited. The theoretical analysis is weak, in that there is no analysis on the convergence and neither how to choose the projection for mirror mapping construction. Experimental results can also be made more convincing, by adding comparisons with bigger datasets, STOA networks, and ablation study to demonstrate why mirror descent is better than proximal gradient descent in this application.